# Sexual Self Discrepancies, Sexual Satisfaction, and Relationship Satisfaction in a Cross-Sectional Sample of Women Who Experience Chronic Vaginal Pain during Sexual Intercourse

**DOI:** 10.3390/healthcare12070798

**Published:** 2024-04-06

**Authors:** Elizabeth Moore, Justin Sitron

**Affiliations:** 1Strategic Impact Solutions LLC, Lafayette, LA 70508, USA; 2Center for Human Sexuality Studies, Interdisciplinary Sexuality Research Collaborative, Widener University, Chester, PA 19013, USA; jasitron@widener.edu

**Keywords:** sexual satisfaction, vaginal pain, self-discrepancies, relationship satisfaction, dyspareunia, sex frequency

## Abstract

One out of three women may suffer from chronic vaginal pain during intercourse, a complex health issue that leads to lasting psychological, sexual, emotional, and relational difficulties even after initial relief. Women who experience this pain condition may compare their sexual selves to the societal norm of being pain-free. Comparisons that do not align with one’s actual sexual self result in sexual self-discrepancies and may cause emotional distress. Sexual self-discrepancies may hinder sexual and relationship satisfaction for women who experience chronic vaginal pain during sexual intercourse. This mixed-method study examined the sexual self-discrepancies women reported and the degree to which their sexual self-discrepancies were related to their sexual and relationship satisfaction. Results from this cross-sectional study showed that the majority of participants experienced sexual self-discrepancies and that they experienced a significant inverse correlation between sexual self-discrepancies and sexual satisfaction. In multivariate models, sex frequency was the strongest predictor of sexual satisfaction. There were no correlations between sexual self-discrepancies and relationship satisfaction. Future measurement research should examine the role of sex frequency in the experience of sexual satisfaction. Education on maximizing pleasure and minimizing pain may aid women to cope with the negative impact of pain.

## 1. Introduction

Nearly 30 million women in the United States suffer from sexual pain disorders, specifically chronic vaginal pain during heterosexual sexual intercourse [1,2,3]. Approximately one in three women will experience chronic vaginal pain during sexual intercourse (CVPDSI) for three or more months at some point in her life [1]. Dargie and Pukall [2] found that women with chronic pain during sexual intercourse experience pain in nearly 80% of their sexual intercourse encounters. When women with chronic pain during sexual intercourse have pain, it is unwanted and persistent, and the pain experience is multifaceted. As a result, women who experience chronic pain during sexual intercourse tend to view their sexual intercourse experience as negative. Women then start to compare their actual sex life with what they thought it should be or want it to be [4]—a discrepancy that makes coping with this condition more complex. Controlled studies have shown that women suffering from the condition report more sexual distress, poorer sexual function, lower sexual satisfaction, and feelings of inadequacy as a sexual partner [5]. Couples experiencing vaginal pain during sexual intercourse report increased intimacy avoidance and greater fear of sustaining their relationship [3,6].

Limited research has been conducted on sexual self-discrepancies, which are cognitive or emotional conflicts arising from comparing oneself to desired or expected versions of oneself. According to Higgins’ self-discrepancy theory [7], a significant misalignment between one’s actual self and ideal or ought selves leads to emotional distress. In the context of sexuality, this manifests as an internal evaluation of one’s actual sexual selves against desired (ideal) or expected (ought) sexual selves. Women experiencing chronic vaginal-penetrative discomfort and sexual pain evaluate themselves against societal standards of consensual and pleasurable sex. When this pain persists, it creates a dissonance between their painful sexual reality and their desired or expected sexual selves, contributing to their overall suffering with limited coping strategies.

In the field of chronic pain, self-discrepancy theory has gained traction for its relevance to pain experience [8,9]. However, exploration of self-discrepancies in women with chronic vaginal pain during sexual intercourse (CVPDSI) remains limited [5]. A sole study by Dewitte et al. [5] investigated ideal sexual self-discrepancies among women with and without CVPDSI, revealing that women with the condition scored lower on measures of actual self and experienced more fear of pain during sex. They also exhibited a stronger inclination to continue intercourse despite feeling sexually inadequate compared to their ideal selves. Women with CVPDSI commonly report various sexual challenges, including lower desire, arousal, lubrication, and satisfaction, as well as communication issues and relationship fears [5,10]. Dewitte et al. [5] observed that these women had lower sexual self-esteem, engaged in less frequent sex, and experienced greater sexual dissatisfaction compared to those without the condition. Given the diagnostic challenges and treatment complexities associated with CVPDSI, examining self-discrepancies may offer insights into women’s emotional regulation based on their perceived and desired sexual identities [5,11].

Sexual satisfaction is integral to the well-being of individuals and their relationships [12]. It encompasses subjective evaluations of positive and negative aspects within one’s sexual relationship [13]. While research has linked satisfying sexual experiences to overall happiness and health [14], experiencing chronic vaginal pain during intercourse adversely affects sexual satisfaction among heterosexual couples [4]. Despite increasing openness to diverse sexual experiences, vaginal-penile intercourse remains central for most heterosexual couples [14]. Thus, it is understandable why women experiencing pain report lower satisfaction. However, sexual satisfaction is nuanced and multifaceted, remaining a crucial aspect of committed relationships [14].

Defining sexual satisfaction involves comparing positive and negative dimensions of sexual experiences [13]. However, measurement strategies vary across studies, leading to inconsistencies. Some studies assess satisfaction through subjective appraisals of one’s sex life, while others focus solely on penile-vaginal intercourse. Additionally, there is a lack of specificity in assessing satisfaction, with few studies exploring particular sexual activities or relationship aspects [15]. Consequently, researchers seek more standardized measures to facilitate meaningful comparisons.

Efforts to develop multidimensional measures of sexual satisfaction include the Interpersonal Exchange Model of Sexual Satisfaction (IEMSS) [13]. Grounded in social exchange theory, the IEMSS evaluates the balance between rewards (positive aspects) and costs (negative aspects) within a sexual relationship. It considers whether individuals receive more rewards than expected, how this balance compares to their expectations, and the perceived equality of rewards and costs compared to their partner’s. These components influence sexual satisfaction, with rewards encompassing pleasurable exchanges and costs encompassing painful or demanding ones [12]. For women with chronic vaginal pain during intercourse, the persistent discomfort may outweigh the rewards of sexual activity, leading to diminished satisfaction.

Regarding measurement, relying solely on orgasm frequency overlooks broader aspects of satisfaction [15]. Women may report high satisfaction despite low orgasm frequency, emphasizing the importance of considering diverse factors. Despite its limitations, the IEMSS offers a comprehensive framework for assessing sexual satisfaction in the context of chronic pain, acknowledging the intricate balance between rewards and costs within sexual relationships.

Sexual satisfaction plays a vital role in intimate relationships, yet its assessment is complex and multifaceted. For women experiencing chronic vaginal pain during intercourse, understanding the interplay between rewards and costs within their sexual relationships is crucial for addressing diminished satisfaction. The IEMSS offers a promising framework for exploring these dynamics, highlighting the need for comprehensive measures that capture the diverse aspects of sexual satisfaction.

Rosen et al. [16] investigated sexual satisfaction in women experiencing chronic pain during intercourse using the IEMSS. They found that both affected women and their partners reported lower sexual satisfaction compared to control couples. Moreover, those experiencing pain reported lower sexual rewards, higher sexual costs, and an imbalanced reward-to-cost ratio, with women bearing the brunt of the pain burden. This suggests that their expectations regarding sexual rewards and costs are not being met. Velten and Margraf [17] identified three significant predictors of sexual satisfaction: sexual function, frequency of sexual activity, and sexual desire discrepancy. Women with chronic pain during sex face challenges in all three areas. Firstly, pain during sex indicates impaired sexual function. Secondly, they tend to avoid sexual activity due to pain. Lastly, they report lower sexual desire, influenced by both actual pain and fear of pain. These discrepancies between their actual experiences and standard expectations may lead to feelings of dissatisfaction, disappointment, inadequacy, and fear, as per Self-Determination Theory (SDT). Further research is needed to explore the relationship between sexual self-discrepancies and potential dissatisfaction, bearing in mind the strong link between sexual and relationship satisfaction.

Sexual satisfaction and relationship satisfaction are pivotal for overall well-being and quality of life within committed relationships [14]. Research consistently shows a positive correlation between sexual and relationship satisfaction [18]. Relationship satisfaction involves weighing positive and negative dimensions of the relationship, influenced by rewards, costs, and comparison levels according to social exchange theory. Women experiencing persistent pain during intercourse, who compare their actual sexual selves to ideal or ought selves, face challenges in achieving satisfaction [6,16]. However, connections between sexual self-discrepancies and satisfaction remain unexplored. Relationship satisfaction also relies on factors like commitment, communication, and mutual decision making [19]. Couples with collaborative communication patterns tend to report higher satisfaction levels, while conflicts and poor communication can diminish satisfaction [20]. Collaborative partnerships foster intimacy, trust, and empathy, enhancing relationship satisfaction [21]. Conversely, lower relationship satisfaction correlates with greater depression symptoms [22]. Expectations play a crucial role in satisfaction research, wherein the perception of receiving rewards or costs influences satisfaction levels. However, the connection between sexual self-discrepancies—how individuals compare themselves to their ideal or expected selves—and satisfaction remains unexplored, especially among women experiencing chronic pain during intercourse. Understanding this association could provide insights into coping mechanisms to alleviate physical and emotional pain, reducing sexual self-discrepancies and enhancing overall comfort until a cure is found.

This study aimed to investigate sexual self-discrepancies in women with chronic vaginal pain during intercourse. It explored the relationships between their actual, ideal, and ought sexual selves to understand the internal struggles associated with chronic vaginal pain. Additionally, it examined the association between sexual self-discrepancies and sexual satisfaction, considering the unique challenges faced by women experiencing chronic pain during sex. This study also investigated the links between sexual self-discrepancies and relationship satisfaction, recognizing the impact of chronic vaginal pain on overall relationship satisfaction levels. Pain intensity and frequency of sex were also assessed for their associations with sexual and relationship satisfaction.

## 2. Materials and Methods

A mixed-methods cross-sectional survey [23] was given to test associations between independent variables (sexual pain severity, frequency of sexual intercourse, and sexual self-discrepancies) and dependent variables (sexual satisfaction and relationship satisfaction) [24]. Given that CVPDSI is a sensitive topic, an anonymous electronic survey was used to allow participants to complete it privately, with the hope that this anonymity would encourage truthful and authentic reporting of sexual pain, actual, ideal, and ought sexual selves, as well as overall sexual and relationship satisfaction.

This study addressed four research questions among women who experience chronic vaginal pain during sexual intercourse:

RQ1. To what degree do women experience sexual discrepancies?

RQ2. What are the ideal and ought sexual self-discrepancy traits of women who experience chronic pain during sex?

RQ3. What is the association between sexual self-discrepancies and sexual satisfaction for women who experience CVPDSI?

RQ4. What is the association between sexual self-discrepancies and relationship satisfaction for women who experience CVPDSI?

Sampling and recruiting for sexuality studies pose more challenges than other research topics due to the taboo nature of sex in the US [25]. Experiencing CVPDSI is often unmentionable due to the shame and embarrassment of sex not being pleasurable for sufferers. Due to the complexity of accurate and accessible medical diagnoses among women who experience CVPDSI, we did not eliminate women who did not have a diagnosis; instead, eligible participants met the following criteria: were 18 or older, identified as cisgender (due to the study’s focus on vaginal anatomy), had engaged in consensual sexual activity with a cisgender man (due to the study’s focus on penile-vaginal intercourse) within the past year, reported experiencing unwanted pain during sexual intercourse at least 75% of the time for a duration of three months or longer, had pain relieved shortly after intercourse, and had the ability to read and write in English. These criteria were in line with delimitations used in the previous pain literature [26,27] to ensure that participants were experiencing chronic and clinically relevant pain.

In order to obtain a substantial number of participants, we conducted an online web-based survey, employing a snowball sampling method to recruit respondents through social media platforms such as Facebook. Given the estimated survey completion time of 15 min, participants were offered the chance to opt into a raffle drawing to randomly be chosen to receive one of four $50 Visa gift cards. This incentive structure mirrored methodologies utilized by previous researchers [28].

The survey was distributed using the Qualtrics online survey platform. The survey instrument included several components, including demographics, questions about their pain during sexual intercourse; questions about their actual, ideal, and ought sexual selves; and questions about their sexual and relationship satisfaction. The Widener University Institutional Review Board (IRB) reviewed the proposal to ensure the protection of participants. The survey took approximately 15 min to complete and comprised 27 items. Respondents took between 9 and 23 min to complete the survey, averaging 12 min. Participants had the option to skip any questions; however, only those who provided complete data on all key variables were included in the final sample. The survey remained open and available until at least 100 fully completed surveys were received. Using the formula 50 + 8M (M = number of independent variables, 4), we determined the minimum sample size (82) to conduct a multiple regression [29]. Data were collected and stored online via Qualtrics on a password-protected computer.

### 2.1. Measures

The survey instrument consisted of several components, including questions about vaginal pain during sexual intercourse, questions about sexual satisfaction, questions about relationship satisfaction, perceived sexual self-discrepancies, and demographic questions. This study used four distinct and unique measures: Vulvar Pain Assessment Questionnaire, Global Measure of Sexual Satisfaction, Global Measure of Relationship Satisfaction, and Sexually Modified Integrated Self-Discrepancy Index.

#### 2.1.1. Demographic Questions

Demographic questions included age, race and ethnicity, sexual orientation, relationship status, and frequency of sex. Demographic questions included multiple response options, with an option to type in a response not listed. Obtaining demographic data allowed us to examine the descriptive characteristics of the sample. To report sexual intercourse frequency, participants selected how frequently they have engaged in penile-vaginal sexual intercourse with their cisgender male partner in the last year from the options of never, once a month, about two to three times a month, once a week, two to three a week, and almost every day.

#### 2.1.2. Vulvar Pain Assessment Questionnaire

The VPAQ [30] was developed to assist in the assessment and diagnosis of vulvar pain. The creators pulled from eight previous established sexual pain scales (Self-Completed Leeds Assessment of Neuropathic Symptoms and Signs; McGill Pain Questionnaire; Pain Quality Assessment Scale; Psychache Scale; Pain Catastrophizing Scale; Vaginal Penetration Cognition Questionnaire; Female Sexual Function Inventory; and Dyadic Adjustment Scale) to create a more efficient comprehensive assessment tool that would capture pain characteristics, emotional/cognitive functioning, physical functioning, coping skills, and partner factors [30].

For this study, we only used the Pain Severity and the Sexual Function Interference Subscales of the VPAQ. An example of a Pain Severity item asks participants to rate the intensity of their vulvar pain on a scale of none, mild, moderate, severe, or worst possible. Another example asks the participant to rate the unpleasantness of her vulvar pain on a scale of none, mild, moderate, severe, or worst possible. There are six items in this subscale. An example of the Sexual Function Interference scale asked participants how much their vulvar pain negatively interferes with their desire for sexual activity on a scale from not at all to very much. These items also have the option to select “I avoid because of pain.” Another example was the following: how much does your vulvar pain negatively interfere with worrying about sexual satisfaction no longer being possible? There are 10 items in this subscale. VPAQ’s original study consisted of good construction and internal consistency [30]. Convergent and discriminant validity were observed when correlated with previously established sexual pain scales (e.g., McGill Pain Questionnaire, Pain Quality Assessment Scale, Psychache Scale, Pain Catastrophizing Scale).

#### 2.1.3. Global Measure of Sexual Satisfaction

The Global Measure of Sexual Satisfaction (GMSEX) is a subscale of the Interpersonal Exchange Model of Sexual Satisfaction [13]. The GMSEX assesses respondents’ satisfaction with their overall sexual relationship. This measure is commonly used in chronic sexual pain studies among partnered women [26]. The GMSEX consists of five items on which participants rate their sexual satisfaction on a 7-point bipolar scale. To remain consistent with the rest of this study’s instruments, the 7-point bipolar scale was modified to 5-point bipolar scale. Modifying a 7-point scale to a 5-point scale increases response rate and response quality while reducing confusion and frustration [31,32]. The first item in the GMSEX asked participants “In general, how would you describe your overall sexual relationship with your partner? For each pair of words below, select the number which best describes your sexual relationship with your partner.” This item asked the participant to rate their sexual relationship from very bad to very good. The second item asked them to rate their sexual relationship from very unpleasant to very pleasant. The third item asked them to rate their sexual relationship from very negative to very positive. The fourth item asked participants to rate their sexual relationship from very unsatisfying to very satisfying. The last item asked participants to rate their sexual relationship from worthless to very valuable. A total Sexual Satisfaction score was derived from averaging the scores on each of the items 1–5. Potential total scale scores range from 1 to 5, with higher scores indicating greater sexual satisfaction.

#### 2.1.4. Global Measure of Relationship Satisfaction

The Global Measure of Relationship Satisfaction (GMREL) is also subscale of the Interpersonal Exchange Model of Sexual Satisfaction [13]. The GMREL assesses respondents’ satisfaction with their overall relationship. This measure is commonly used in chronic sexual pain studies among partnered women when measuring relationship satisfaction [26]. The GMREL asked participants to rate their overall relationship with their partner on a 7-point bipolar scale. To remain consistent with the other instruments in this study, the 7-point bipolar scale was modified to a 5-point bipolar scale. Modifying a 7-point scale to a 5-point scale increases response rate and response quality while reducing confusion and frustration [31,32].

The first item asked participants “In general, how would you describe your overall relationship with your partner? For each pair of words below, select the number which best describes your relationship with your partner”. This item asked participants to rate their relationship from very good to very bad. The second item asked them to rate their relationship from very pleasant to very unpleasant. The third item asked them to rate their relationship from very positive to very negative. The fourth item asked participants to rate their relationship from very satisfying to very unsatisfying. The last item asked participants to rate their relationship from very valuable to worthless. Relationship Satisfaction score was derived from averaging the scores on each of the items 1–5. Potential total scale scores range from 1 to 5, with higher scores indicating greater relationship satisfaction.

#### 2.1.5. Sexually Modified Integrated Self-Discrepancy Index

This portion of the survey comprised self-discrepancies relating to participants’ sexual self and sexual ideals and sexual expectations [33,34]. There are no validated measurements of sexual self-discrepancy that have been replicated [5], and numerous researchers have modified self-discrepancy measurements to include the sexual self while maintaining internal consistency [5,34,35,36]. Thus, the Integrated Self-Discrepancy Index (ISDI) was modified to focus on the sexual self rather than the entire self.

To ensure participants responded specific to their sexual selves, a statement was presented in the instructions before the scale “Please answer the following questions regarding your current sexual self particularly when you are soon to engage or engaging in sexual intercourse with your partner?” The inclusion of this statement about their sexual self while having sexual intercourse encouraged participants to consider their experience of sexuality when completing the ISDI [37]. The original ISDI asked participants to list five traits or attributes for their ideal self; however, we focused on their ideal sexual self. Participants were asked to list five traits they would ideally like to possess sexually. This list was their ideal sexual self list. Once they created their ideal sexual self-trait list, they were asked to individually rate how much they think each of their self-reported ideal sexual self-traits described their current sexual self, with responses ranging from 1 (completely describes me) to 5 (does not describe me). The average of all five ideal sexual self-scores was their ideal sexual self-discrepancy score. Higher ideal sexual self-discrepancy scores indicated higher ideal sexual self-discrepancies, suggesting the participant’s actual sexual self was not congruent with who they would ideally like to be sexually (ideal sexual self). Lower ideal sexual self-discrepancy scores indicated minimal ideal sexual self-discrepancies, suggesting the participant’s actual sexual self was comparable with who they ideally want to be sexually.

Additionally, the original ISDI asked participants to list five traits or attributes for their ought self; however, we focused on their ought sexual self. Participants were asked to list five traits they should possess sexually. This list was their ought sexual self list. Once they created their ought sexual self-trait list, they were asked to individually rate how much they think each of their self-reported ought sexual traits describes their current sexual self, with responses ranging from 1 (completely describes me) to 5 (does not describe me). The average of all five ought sexual self-item scores is their ought sexual self-discrepancy score. Higher ought sexual self-discrepancy scores indicated larger ought sexual self-discrepancies, suggesting the participant’s actual sexual self was different from the sexual self they think they should be (ought sexual self). Lower ought sexual self-discrepancy scores indicated minimal ought sexual self-discrepancies, suggesting the participant’s actual sexual self was very similar to their ought sexual self. If participants were unable to list five traits or attributes for each sexual self-state, they were shown 45 sexual-related adjectives reported in prior research [38] to choose from; providing a list of adjectives is standard practice in the use of the ISDI if participants have a hard time thinking of traits or attributes [39]. Having participants think of their traits or attributes and list them from memory is common practice in self-discrepancy research [7,40]. Traits or attributes listed by the participants were analyzed by the researcher using inductive coding. Inductive coding consists of reading the raw textual data to develop common themes [41]. The trait themes will be discussed further in RQ1 analyses.

### 2.2. Data Analysis

We ran these processes, and the data tables below show the results. We used content analysis to examine these sexual self traits, allowing researchers to consolidate extensive textual data into a finite set of categories, facilitating efficient analysis. Words were recoded manually if they came in different forms, but traits were very similar. For example, “painfree”, “pain free”, and “no pain” were all categorized as “pain free”. Particular discrepant sexual self-traits were then counted and reported. General trends and patterns were then identified [42]. For reliability purposes of content analysis, the same coding process was completed a second time one month after the initial coding process was complete.

Bivariate Pearson Correlation Coefficient (r) analyses were employed to examine the association between independent (sexual pain severity, frequency of sexual intercourse, and sexual self-discrepancies) and dependent variables (sexual satisfaction and relationship satisfaction) to determine the presence of a statistically significant linear relationship between variables [43]. The interpretation of correlation strengths was guided by effect size conventions, which categorize numeric values into qualitative indicators ranging from small to huge. A correlation coefficient (r) squared value of 0.14 was considered to indicate a large effect size. Multiple linear regression was conducted using independent variables and predictors combined to determine the variability in the respective dependent variable, sexual satisfaction or relationship satisfaction. The degree of sexual pain and frequency of penile-vaginal sexual intercourse were added as predictors to determine if there was a relationship between these covariates and the respective dependent variables. The assumptions of homoscedasticity and linearity were assessed through the visual inspection of scatter plots. The assumption of multicollinearity was considered satisfied if tolerance values did not exceed 0.10. Durbin–Watson values between 1.0 and 3.0 were considered as satisfying the assumption of Independence of Error. The assumption of normality of residuals was considered as satisfied if skew values did not exceed −2.0/+2.0 and kurtosis values did not exceed −7.0/+7.0. Cook’s Distance values below 1.0 were deemed suggestive of meeting the assumption of insignificance regarding outliers.

## 3. Results

A total of 106 women who experience CVPDSI formed the sample in this study, see Table 1. Women ranged in age from 19 to 50 years (M = 27; SD = 6.40). Nearly 39% of the women were married, 6% were engaged, and nearly 50% were dating one person exclusively. More than two-thirds of the participants (69%) identified as heterosexual, and 20% identified as bisexual. A majority (84%) of participants identified their race/ethnicity as white. Half of the women (52.9%) have been dealing with vulvar pain for more than five years; 44.3% of participants have been dealing with this pain for five years or less. Nearly all of the participants (95%) indicated they have discussed their pain during sexual intercourse with their physicians. Of the participants who have talked to their physician, 75% received a diagnosis of vulvodynia, vestibular vulvodynia, pelvic floor dysfunction, and/or vaginismus. Of the entire sample, 105 participants reported how often they attempted sexual intercourse in the last month. Results indicate that in the past month, a majority of the participants reported either never having sex (30.2%) or attempting sex about once a month (32.1%). The internal reliability for sexual (a = 0.91) and relationship satisfaction (a = 0.96) were excellent.

### 3.1. Initial Descriptive Statistical Findings

The descriptive statistical analyses focused on evaluations based on item frequency (*n*) typicality (mean scores), variability (minimum/maximum; standard deviations), standard error of the mean scores (SEM), and normality of data arrays (skew; kurtosis). Table 2 contains a summary of findings for the descriptive statistical analyses. Participants reported their level of average pain intensity ranging from 2.17 to 5.00 (M = 3.66, SD = 0.59), with higher scores indicating higher levels of pain intensity. Nearly one-third of participants indicated having severe to worst possible pain on average, 58% indicated having moderate pain, and 10% indicated having mild pain intensity on average. The mean sexual pain score of 3.66 on a scale of 2.17–5 indicates that most respondents experienced moderate to high levels of consistent pain.

### 3.2. Findings by Research Question

Descriptive and inferential statistical techniques were used in the analysis of this study’s four research questions. The probability level (alpha) for a finding to be considered statistically significant was established at *p* ≤ 0.05. The qualitative descriptions of effect sizes achieved in the study were stated in accordance with the conventions of interpretations proposed by Cohen [44].

#### 3.2.1. To What Degree Do Women Experience Sexual Discrepancies?

Nearly 98% of the participants experienced sexual self-discrepancy (*n* = 104). For the ideal sexual self-discrepancy scale, the average of all five ideal sexual self-scores was their ideal sexual self-discrepancy score, with higher scores indicating larger ideal sexual self-discrepancies. The ideal sexual self-discrepancy scale had a possible range from 1 to 5 (M = 3.89, SD = 0.92). The average of 3.89 indicates that participants had a moderate level of ideal sexual self-discrepancy. Very few participants (*n* = 5) reported traits that mostly or completely described them, indicating that these individuals believe they possess their ideal sexual traits. Few participants (*n* = 8) reported that the listed ideal sexual trait describes them very well, indicating these participants are low ideal sexual self-discrepant. A quarter of the participants (*n* = 31) indicated that their list of ideal sexual traits described them moderately well. Nearly half of participants (*n* = 47) indicated that their list of ideal sexual traits described them slightly well, suggesting that these participants are high ideal sexual self-discrepant and have discrepant views between what they ideally want to be sexually and how they actually are sexually. Of the entire sample, 15 participants indicated that all five of their ideal sexual traits did not describe them at all, suggesting that these participants do not embody the sexual traits they want to possess.

The average of all five ought sexual self-scores was calculated to determine participants’ ought sexual self-discrepancy scores, with higher scores indicating larger ought sexual self-discrepancies. The ought sexual self-discrepancy scale had a possible range from 1 to 5 (M = 3.80, SD = 1.00). The average of 3.80 indicates that participants had a moderate level of ought sexual self-discrepancy. Only eight participants reported traits that mostly or completely described them, indicating that these individuals believe they possess sexual traits they should possess. Nine participants reported that the listed sexual trait describes them very well. Nearly one-third of the participants (*n* = 34) indicated that their list of ought sexual traits described them moderately well. Nearly half of participants (*n* = 42) indicated that their list of ought sexual traits described them slightly well or not at all. Of the entire sample, 11 participants indicated that all five of their ought sexual traits did not describe them at all, suggesting they do not embody any sexual traits they believe they should possess.

#### 3.2.2. What Are the Ideal and Ought Sexual Self-Discrepancy Traits of Women Who Experience Chronic Pain during Sex?

Participants were asked to write down five traits they wanted to possess and five traits they believe they should possess. If participants were not able to think of five traits, they were able to look at a list of sexual traits to use. In total, there were 62 ought sexual traits and 93 ideal sexual traits listed by participants. Content analysis was used due to its ability to identify themes from a set of qualitative data. Using an inductive approach to coding, numerous ideal and ought sexual traits were explored. Initially, an examination was conducted on the frequency of traits listed by participants. Following this, the mean discrepancy score for each of the identified traits was analyzed. Exploring traits that were most discrepant provided insight into what sexual traits participants believe they should or want to possess but do not, creating the most self-discrepant sexual traits in the study. Table 3 includes ought and ideal sexual traits self-reported the most by participants. This table displays the sexual traits participants thought of the most when asked to think of ought or ideal sexual traits. Additionally, the discrepancy scores are listed based on how discrepant respondents who listed the trait reported being different than their current sexual self.

Ought Sexual Self-Discrepant Trait Themes. Ought sexual self-discrepant traits are traits that participants believe they should possess sexually, but they have indicated that these traits do not describe them at all. Pain-Free was listed the most by participants (*n* = 36) and had a large discrepancy average (M = 4.91). Carefree was listed by 14 participants as a trait that does not describe them at all. Spontaneous and Adventurous were also listed often and had a higher discrepancy score. Frequent was the most ought sexual self-discrepant trait listed; however it was only listed by five participants.

Ideal Sexual Self-Discrepant Trait Themes. Ideal sexual self-discrepant traits are those that participants ideally want to possess sexually, but they indicate that this trait does not describe them at all. Pain-free and painless were terms that arose among 48 participants who indicated the traits do not describe them at all. These traits had the largest amount of sexual self-discrepancy for participants. Carefree was listed for 24 participants as highly discrepant of their actual sex life. A total of 15 participants listed Spontaneous as being an ideal sexual self-discrepant trait. Enjoyable and Pleasurable were listed for 15 participants. Other themes included Confidence (*n* = 13), Healthy (*n* = 10), Guilt Free (*n* = 6), and shameless (*n* = 2).

The traits listed the most for ideal were slightly different than the traits for ought. The most common term across both categories was Pain-Free. Pain-Free was listed by 59 women for ideal; however, only 36 individuals listed it for ought sexual traits. This may be due to misinformation or social conditioning that sex is expected to be a little painful. Additionally, ideal sexual traits focus on pleasure and being fun and playful, whereas ought sexual traits focus more so around being loving and passionate.

#### 3.2.3. What Is the Degree of Association between Sexual Self-Discrepancies and Sexual Satisfaction for Women Who Experience CVPDSI?

Ought Self-Discrepancies and Sexual Satisfaction. In a Pearson correlation analysis, there was a significant inverse correlation between ought sexual self-discrepancies and sexual satisfaction, r(106) = −0.361, *p* < 0.001. This suggests that as ought sexual self-discrepancies increase, sexual satisfaction decreases. Figure 1 presents a scatterplot of the relationship between the two variables.

Ideal Sexual Self-Discrepancies and Sexual Satisfaction. Pearson Product–Moment Correlation (r) results indicated there was a significant inverse correlation between ideal sexual self-discrepancies and sexual satisfaction, r(106) = −0.370, *p* < 0.001. This suggests that as ideal sexual self-discrepancies increase, sexual satisfaction decreases. Figure 2 presents a scatterplot of the relationship between the two variables.

Multivariate Analysis. A multiple linear regression analysis was conducted to assess the degree to which ideal sexual self-discrepancy and ought sexual self-discrepancy, pain intensity, and frequency of sex were predictive of perceptions of overall sexual satisfaction. Dewitte et al. [5] found that women who experienced pain during sex reported less frequent sex and lower sexual satisfaction than women who did not have this condition. Sex frequency is one of the predictors for higher sexual satisfaction for women in heterosexual relationships [13]. However, due to the direct pain that is only associated with sexual intercourse for women who experience CVPDSI, we wanted to explore both variables in a multivariate model with ideal and ought sexual self-discrepancy scores.

Prior to conducting multivariate linear regression analysis, we assessed for the assumptions of linear regression analyses, each of which the data met. First, we used partial regression plots to assess linearity and a plot of Studentized residuals against the predicted values. As assessed by a Durbin–Watson statistic of 2.308, there was independence of residuals. We found homoscedasticity by visually inspecting a plot of Studentized residuals compared to unstandardized predicted values. There was no multicollinearity, which we assessed by tolerance values greater than 0.1. There were no Studentized deleted residuals greater than ±3 standard deviations, no leverage values greater than 0.2, and values for Cook’s distance were above 1. The assumption of normality was met given the data closely following the normality trend line, as assessed by P-P Plot. The predictive model was statistically significant (*F*(4, 100) = 11.86, *p* < 0.001, R^2^ = 0.32), indicating that approximately 32.2% of the variance in overall sexual satisfaction is explainable by ideal sexual self-discrepancy, ought sexual self-discrepancy, pain intensity, and sex frequency. While pain intensity (*B* = −0.33, *p* < 0.05) and sex frequency (*B* = 0.243, *p* < 0.001) contributed significantly to the model, ought sexual self-discrepancy (*B* = −0.14, *p* = 0.176) and ideal sexual self-discrepancy did not (*B* = −0.214, *p* = 0.057). For pain intensity, on average, a one-unit increase in pain intensity will decrease sexual satisfaction by 0.33 units. However, on average, a one-unit increase in sex frequency will increase sexual satisfaction by 0.24 units. Overall, frequency of sex was the strongest predictor of sexual satisfaction, suggesting that as frequency of sex increases, sexual satisfaction also increases. Table 4 presents the findings of the multiple linear regression.

#### 3.2.4. What Is the Degree of Association between Sexual Self-Discrepancies and Relationship Satisfaction for Women Who Experience CVPDSI?

Ought Self-Discrepancies and Relationship Satisfaction. Pearson Product–Moment Correlation (r) analysis indicated there was not a significant correlation between ought sexual self-discrepancies and relationship satisfaction, r(106) = −0.11, *p* = 0.281.

Ideal Sexual Self-Discrepancies and Relationship Satisfaction. Pearson Product–Moment Correlation (r) indicated there was not a significant correlation between ideal sexual self-discrepancies and relationship satisfaction, r(106) = −0.05, *p* = 0.646.

Multivariate Analysis. Despite the non-significant bivariate relationships, as established with the Pearson correlations, a multiple linear regression was conducted to assess the degree to which ideal sexual self-discrepancy and ought sexual self-discrepancy, pain intensity, and frequency of sex were predictive of perceptions of overall relationship satisfaction. The predictive model was not statistically significant (F(4, 100) = 0.84, *p* = 0.504, R^2^ = 0.032). None of the predictors significantly predicted relationship satisfaction.

## 4. Discussion

Most participants experienced moderate to severe pain when engaging in sexual intercourse, similar to other research on sexual pain in women [45]. Most of the participants indicated that they were sexually self-discrepant with the sexual traits they selected in the survey. The most common theme for both ideal and ought sexual self-discrepant traits was Pain-Free, which supports Pâquet et al. [4], who found couples who experience genito-pelvic pain during sex believed in the idea of pain-free sex. This finding was not surprising since CVPDSI involves unwanted pain.

Results from the Pearson correlation in the present study indicated a significant correlation for both ought and ideal sexual self-discrepancies and sexual satisfaction. There was a stronger correlation between ideal sexual self-discrepancies and sexual satisfaction. There was no correlation between sexual self-discrepancies and relationship satisfaction.

When adding frequency of sex and pain intensity variables, frequency of sex had the most influence on overall sexual satisfaction, whereas sexual self-discrepancy was no longer significant. Although these findings suggest that women experience significant sexual self-discrepancies and that these discrepancies may impact their sexual satisfaction, the number of times a person had sex had more influence on sexual satisfaction. There was no relationship found among sexual self-discrepancies, pain intensity, and sex frequency on relationship satisfaction. The sections below include discussion of the findings in relation to the previous literature.

Findings from this study show that women experience sexual self-discrepancies, indicating that women compare themselves to traits that are not congruent with their actual selves. Similar to Dewitte et al. [5], most women experienced large discrepancies between actual and ideal sexual traits. Participants, on average, reported having higher ideal sexual self-discrepancies than ought sexual self-discrepancies. Most women also experienced large discrepancies between actual sexual traits and sexual traits they believe they should possess. The findings suggest that women who have CVPDSI experience major discrepancies. The variance between ideal and expected sexual standards might stem from the intricate and multifaceted nature of human sexuality. This complexity encompasses diverse societal norms and individual desires regarding sexual behaviors, traits, roles, and culturally acceptable dynamics, highlighting the contrast between perceived ideals and perceived expectations [46]. Researchers and practitioners need to investigate women’s sexual self-discrepancies and help close the gap between what women believe they should be, what they ideally want to be, and who they currently are.

The trait that was identified the most in this sample was Pain-Free. Interestingly though, Pain-Free was only identified 36 times for ought sexual traits, whereas for ideal sexual traits, Pain-Free was identified 59 times. This may be due to women believing or hearing from others that sex should hurt [47,48,49]. Carter et al. [50] conducted a study that included 2007 individuals who experienced pain during the past year, and they found that many women normalized the pain and viewed it as inevitable because it was a long-lasting issue. Many women report their first sexual encounters as painful [50] and may overlook the pain because there may be accompanying positive feelings of sex (e.g., arousal, pleasure, connection, intimacy). However, there is growing awareness of sexual functions, sexual arousal, and the possibility of pleasure for women. It was not until recently that even scholars acknowledged the importance of sexual pleasure. The World Association of Sexual Health recently adopted sexual pleasure as being an important component of sexual health [49]. All of this combined may give us an answer as to why over half of the women reported Pain-Free as an ideal sexual trait, whereas only one-third of the women reported Pain-Free as an ought sexual trait.

There were many traits that were identified on both ideal and ought lists. Outside of the Pain-Free trait, ideal sexual traits were more focused on pleasure and fun, while ought sexual traits were more focused on loving and passion. Furthermore, when looking at the most highly discrepant traits, the most discrepant ought sexual traits were Pain-Free, Frequent, Carefree, Easy, Spontaneous, Adventurous, and Enjoyable. These traits align with the notion that women “should be” ready and willing to have sex at any given time. Current research on gender norms and expectations concerning heterosexual sex holds that women are passive and say yes to sex while men are entitled to want and ask for sex [51]. Women who have CVPDSI have an additional barrier to being “ready” and “willing”, thus creating a larger gap between what is expected of them and what they are actually experiencing.

The highest ideal sexual discrepant traits were Guilt-Free, Pain-Free, Spontaneous, Enjoyable, Playful, and Amazing. Women want to have a fun-filled sexual experience. However, if women do not have information on how to make fun and enjoyable sexual experiences coincide with the pain condition, women are left with a sexual discrepancy. Future research should explore the meaning of these traits to women and their contexts. For example, Guilt-Free was listed as being one of the discrepant sexual traits. Does this guilt come from feeling guilty over having sex in general [52,53] or does it relate to feeling guilty about having this pain condition [54], or perhaps something else? Mosher and Cross [55,56] conceptualized sex guilt as “generalized expectancy for self-mediated punishment for violating or for anticipating violating standards of proper sexual conduct”. Azim et al. explored the prevalence of painful sex in adolescents and young women as it relates to sex practices, religiosity, gender role beliefs, sex education, and sex guilt and found that sex guilt was significantly associated with painful sex and that religiosity may be one avenue through which women can experience sexual guilt.

Therapists and medical providers could examine patients’ sexual self-discrepancies to aid in interventions that help individuals seek more realistic sexual traits so that the discrepancy would not be so large or impossible to attain. Awareness of these traits may also help in coping with their ideas of what they should be or want to be. Nonetheless, traits that were less discrepant were healthy relational traits such as Pressure-Free, Loving, and Caring. It is reassuring that although this condition may cause sexual self-discrepancies, women experience positive aspects of their sexual selves.

### 4.1. Sexual Satisfaction

This study found a significant inverse correlation between ought and ideal sexual self-discrepancies and sexual satisfaction. For women who do not align with traits they believe they should have or want to have, their sexual satisfaction decreases. Although there was a significant correlation between sexual self-discrepancies and sexual satisfaction, when variables were combined into a regression model, ideal and ought self-discrepancy were not a predictor of sexual satisfaction. Frequency of sex and pain intensity were predictors of sexual satisfaction.

Surprisingly, frequency of sex was the largest positive predictor of sexual satisfaction; as the amount of sex increased, so did sexual satisfaction. One may think this would not be the case for women who experience CVPDSI. Furthermore, one may think that more frequent attempts at vaginal-penile sex would be associated with lower sexual satisfaction because of the pain felt during penetration. However, this finding does not support that claim and suggests that although sexual intercourse may involve vaginal pain, the frequency of sexual intercourse positively influences sexual satisfaction. Velten and Margraf argued that frequency of sex is one of three significant predictors of sexual satisfaction for women in general, and the present study supports that claim, even among women who experience pain during intercourse. This may be due to sexual satisfaction being complex and involving numerous interpersonal, physiological, relational, and social factors such as intimacy, not necessarily the physical act or physical feeling of sex alone [13,17,57], as discussed further below.

Frequency of sex being the largest predictor of sexual satisfaction also calls into question the construct validity of the sexual satisfaction measure, as it may not be capturing, or may not be sensitive to, the ways in which pain affects satisfaction. It may also be that people who engage in sex more frequently make increased use of pain-reduction methods [58] such as shallower insertion or using more lubrication. Likewise, they may seek out sex for intimacy rather than for physical pleasure. Bergeron et al. [59] found that couples who maintain adequate levels of intimacy may be better able to cope with the consequences of the pain, which may enhance sexual satisfaction.

### 4.2. Relationship Satisfaction

Results showed no association between ought or ideal discrepancies and relationship satisfaction, suggesting that although women experience a discrepancy between who they are and who they want to be or should be, this discrepancy does not influence relationship satisfaction. The findings of previous research on sexual pain and sexual satisfaction and relationship satisfaction, however, are mixed. Although pain interferes with a woman’s sex life and sexual satisfaction, research shows that this condition does not always make a significant difference in relationship satisfaction [59]. Other researchers, however, have found that women and couples who experience CVPDSI report lower relationship satisfaction than those who do not experience it [6,16].

It is interesting that sexual self-discrepancies had a significant association with sexual satisfaction but not relationship satisfaction. However, it makes sense that sexual self-discrepancies (who I am compared to, who I want to be and should be sexually) do not play a significant role in relationship satisfaction. These intrapersonal aspects may be minimal compared to other factors (e.g., communication, trust, respect, connection) that largely influence relationship satisfaction. This is reassuring because women who experience CVPDSI report fear of losing the relationship [4]. Sexual self-discrepancies can feel detrimental to one’s relationship because one is not who they believe they should be or want to be sexually. In heterosexual relationships, sexual expectations and norms around sexual intercourse feel heavier when someone is not living up to them because sexual intercourse is the main sexual experience in heterosexual relationships. Knowing that one’s sexual self-discrepancy does not play a significant role in relationship satisfaction brings a certain level of peace.

### 4.3. Implications and Recommendations for Practice

This study revealed a significant association between sex frequency and sexual satisfaction, suggesting practical interventions to alleviate pain and enhance pleasure among women with chronic vaginal pain during sexual intercourse (CVPDSI). By addressing pain reduction and sexual self-discrepancies, practitioners can potentially improve sexual experiences and satisfaction for affected individuals. The following implications and recommendations are based on established best practices in the field, drawing from a synthesis of existing research and expert consensus.

#### 4.3.1. Minimizing Pain 

Addressing Sexual Self-Discrepancies: Many women with CVPDSI experience moderate to high sexual self-discrepancies, comparing themselves to unattainable traits such as Pain-Free, Easy, and Spontaneous. It is essential to inform these women that the condition is not curable and may persist. Educating women about the condition’s permanence can foster acceptance, encouraging more realistic ideal traits and minimizing the discrepancy gap. Interventions targeting self-discrepancies have shown promise in improving quality of life and reducing negative emotions associated with the condition.

Feminist Perspectives and Questioning Ideal Traits: Encouraging women to question the origin of their ideal and ought sexual self traits, particularly in the context of patriarchal norms, can empower them to challenge unrealistic expectations. By unlearning unattainable traits rooted in societal norms, women can replace them with obtainable and self-centered traits, promoting sexual satisfaction and self-acceptance.

Healthcare Practices: Medical providers should proactively screen for CVPDSI and pelvic pain disorders, as many women may not report symptoms unprompted due to social stigma. Open communication about the condition’s effects and available treatments is crucial. Lidocaine application, mindfulness-based therapy, and cognitive-behavioral therapy have shown efficacy in reducing pain and enhancing sexual function. Multidisciplinary approaches, though challenging to implement, have demonstrated significant improvements in pain and distress.

Education on Pain Condition: Education could be the most effective and cost-efficient way to support women who experience CVPDSI. There is compelling evidence that educational programming or conversations have been associated with reduced pain, disability, anxiety, and stress for chronic pain conditions [17]. Education can also help reduce sexual and psychological distress and increase pleasure for women who have dyspareunia [17,60]. Education could inform individuals that the condition is a real, diagnosable health condition. Education could aid in women’s ability to accept this pain condition and inform the individual that this condition will not go away on its own. Acceptance of the pain condition has been found to be associated with better functioning and psychological outcomes [61]. Boerner and Rosen [62] found that greater pain acceptance was associated with lower self-reported pain during sex, lower anxiety and depression, greater sexual functioning, and greater sexual satisfaction for the individual and their partner. More information is needed on how catastrophizing, a negative cognitive response to anticipated pain, and coping mechanisms are related to women being anxious or relaxing during intercourse, resulting in increased or reduced pain, respectively. For women who experience sexual self-discrepancies, acceptance of this pain condition may help reduce the discrepancies between wanting to be pain-free and having painful experiences. Acceptance may also help minimize the distance between their actual sexual selves and aid in having more achievable ought and ideal sexual traits.

Support: Ghai et al. [63] also found that support from other women or a support network reduced feelings of isolation and allowed women to discuss their attitudes and anxiety over the condition, relational side effects, and empathy. Having a support network empowers women to discuss their difficulties, whereas communicating their pain to others may cause embarrassment. For this reason, medical professionals should also advise that there are social media support networks that have thousands of users and groups for partners of women who have this condition. Nonetheless, for women who experience CVPDSI and sexual self-discrepancies, pleasure and pain can co-exist. Minimizing pain while maximizing on pleasure can give women a new realization that having this condition gives permission to explore what pleasure is and in what ways.

#### 4.3.2. Maximizing Pleasure

Genital Vibration/Vibrator: Vibrations have shown therapeutic benefits for sexual dysfunctions, pelvic floor dysfunction, and vulvar pain [64], enhancing sexual arousal and overall function. Clinicians increasingly recommend vibrators as therapy. Health providers, therapists, and sexuality educators should inform women experiencing sexual pain about vibrator benefits, using alternative terms like “sex toy” or “personal massager” depending on patient comfort. Nearly half of women and men have used vibrators for sexual enhancement, destigmatizing their use. Non-penetrative vibrator use can maximize pleasure while minimizing pain through nerve stimulation. Encouraging exploration of vibrators beyond genital use and involving partners is beneficial. Practitioners should facilitate conversations about introducing vibrators to partners. Vibrators are available online or in adult retail stores, and mentioning lubricant use can further enhance pleasure.

Sensate Focus Technique: Sensate focus exercises promote body exploration and intimacy, reducing stress and anxiety associated with sex [5]. These exercises encourage sensual and sexual mindfulness, fostering communication and familiarity with partners’ bodies and erogenous zones. Communication during exercises enhances understanding of sensual and sexual pleasures, assisting individuals in coping with pain and exploring pleasurable sexual traits.

Communication with Significant Other: Couples affected by sexual pain may have lower sexual communication quality, impacting sexual and relational outcomes negatively [57]. Greater sexual communication correlates with higher sexual satisfaction, function, and lower depressive symptoms. Collaborative problem solving and communication can address partner expectations, alleviating some ought self-discrepancies.

Sexual Intimacy: Greater affection outside sexual intercourse correlates with higher sexual satisfaction, acting as a protective buffer against emotional distress [57]. Sex frequency predicts sexual satisfaction, possibly attributed to sexual intimacy during intercourse. Intimacy encompasses closeness, pleasure, trust, and acceptance, and while it may not differ significantly between women with and without painful sex experiences, it may indirectly mitigate the pain condition. Further research is warranted on intimacy’s role in sexual satisfaction for women with chronic vaginal pain during intercourse.

### 4.4. Limitations

The sample for this study is restricted to cisgender heterosexual women in relationships with cisgender men, lacking a control group for comparison with women not experiencing chronic vaginal pain during sexual intercourse (CVPDSI). Furthermore, the absence of random sampling limits the generalizability of findings to the broader female population. Additionally, most participants were white heterosexual women, which restricts the applicability of the results to lesbian or queer women, transgender individuals, or women of color. This underrepresentation of people of color may result from the underreporting or neglect of pain experiences among this demographic, highlighting the need for further research within communities of color and queer communities.

Moreover, online survey administration may inadvertently exclude women without internet access or technical proficiency, contributing to potential sampling bias. Furthermore, the sensitive nature of discussing CVPDSI may introduce social desirability bias, wherein respondents provide socially acceptable responses rather than expressing their true feelings. Consequently, there may be underreporting of undesirable activities, especially regarding sexual behaviors.

Measurement errors are also a concern, given the adaptation of the Sexual Self-Discrepancy Index (ISDI) for this study. While the ISDI appeared suitable for measuring sexual self-discrepancies, this area of research is relatively new, with limited prior studies. Moreover, the specific scale used in this study has not been previously employed in similar research. Additionally, the scale’s reliance on participants’ own sexual attributes or traits, supplemented by a predefined list, may not encompass all possible attributes, potentially influencing measurement accuracy.

### 4.5. Recommendations for Future Research

Given the prevalence of sexual self-discrepancies among women, future research should explore this phenomenon through both qualitative and quantitative methodologies to obtain comprehensive insights. Qualitative inquiries could delve into participants’ identification of sexual traits, their motivations behind desiring or believing in those traits, and the alignment of their actual selves with these traits. Such investigations would inform educators and practitioners in guiding women towards redefining unrealistic sexual expectations with more attainable ones. Additionally, further research should explore sexual self-discrepancies in greater depth.

Regarding the measurement of sexual satisfaction, it is essential to reconsider the appropriateness of existing scales, particularly for individuals experiencing pain during intercourse. The Global Measure of Sexual Satisfaction (GMSEX), commonly used in such studies, may not adequately capture the experiences of this population due to its limited scope. Utilizing comprehensive measures such as the Inventory of Interpersonal Problems—Circumplex Scales (IIP-C), which assesses the costs and rewards of relationships more extensively, could provide a more nuanced understanding of the factors influencing sexual satisfaction in this context.

Further investigation is warranted into the specific aspects of sexual satisfaction referenced by individuals experiencing pain during intercourse. McClelland’s [15] Intimate Justice framework offers a valuable theoretical lens, examining how social and political inequities impact intimate experiences. This perspective considers proximal and distal factors affecting sexual and relational well-being, including freedom from harm, experiences of pleasure, and the influence of social norms. Researchers could use this framework to explore women’s motivations for engaging in sex, distinguishing between pleasure-driven participation and obligatory engagement, thereby enhancing our understanding of sexual satisfaction among this demographic.

## 5. Conclusions

Sexual self-discrepancies are apparent in women who experience chronic vaginal pain during sexual intercourse. Greater sexual self-discrepancies are strongly associated with lower sexual satisfaction and are not associated with relationship satisfaction. Most importantly, sex frequency was associated with greater sexual satisfaction despite having this pain condition. More research is needed to explore the construct validity of sexual satisfaction measures for people who have sexual pain disorders to better understand what this population is thinking of when they respond to questions about their sexual satisfaction. Furthermore, more research is needed to investigate sexual self-discrepancies in general and within sexual pain disorders. For women in intimate relationships who experience this condition, education on sexual self-discrepancies, maximizing pleasure, and minimizing pain may aid in coping with the negative effects until we find a cost-efficient and effective treatment.

## Figures and Tables

**Figure 1 healthcare-12-00798-f001:**
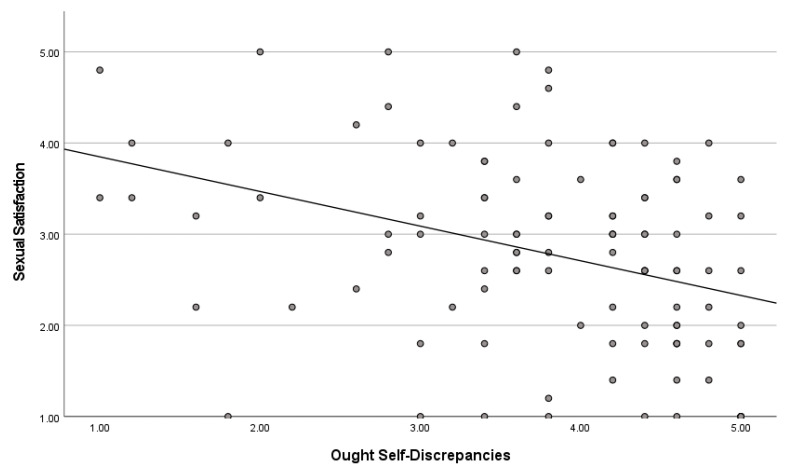
Scatterplot between ought self-discrepancies and sexual satisfaction.

**Figure 2 healthcare-12-00798-f002:**
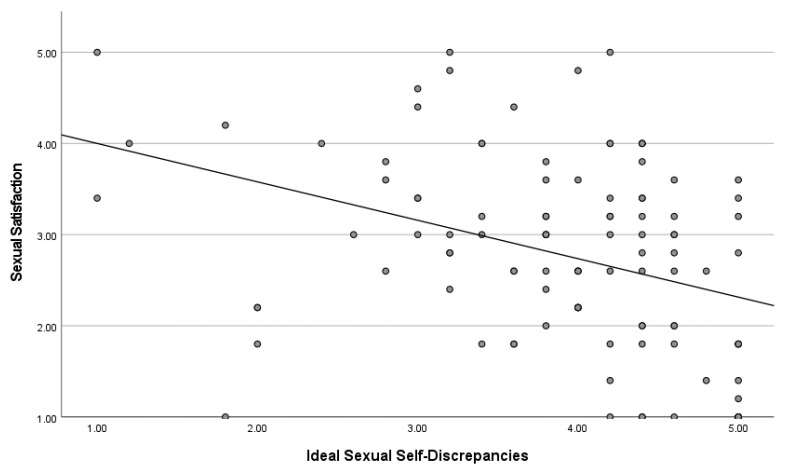
Scatterplot between ideal self-discrepancies and sexual satisfaction.

**Table 1 healthcare-12-00798-t001:** Demographic characteristics.

Characteristic	*n*	%
Relationship Status		
Married/domestic partnership	41	38.7
Engaged	7	6.6
Dating one person exclusively, living together	18	17
Dating one person exclusively but not living together	29	27.4
Dating more than one person	6	5.7
In relationship with more than one long term partner	1	0.9
Sexual Orientation		
Straight or heterosexual	73	68.9
Bisexual	21	19.8
Pansexual	6	5.8
Queer	3	2.8
A sexual orientation is not listed	1	0.9
Sex Frequency		
Never in the past month	32	30.2
Attempted once in the past month	34	32.1
Two or three times in the past month	18	17.0
Once a week	10	9.4
Two to three times a week	10	9.4
Once a day	1	0.9
Education Level		
Less than high school	1	0.9
High school graduate	7	6.6
Some college	12	11.3
2-year degree (associate’s or trade school)	9	8.5
4-year degree (bachelor’s)	43	40.6
Professional degree (master’s)	28	26.9
Doctorate	4	3.8

**Table 2 healthcare-12-00798-t002:** Descriptive statistics: initial findings.

Variable	M	SD	SE_M_	Min	Max	Skewness	Kurtosis
Pain Intensity	3.66	0.59	0.05	2.17	5.00	0.08	−0.08
Sexual Satisfaction	2.78	1.05	0.09	1.00	5.00	0.28	−0.62
Relational Satisfaction	3.79	1.11	0.10	1.00	5.00	−0.95	0.29
Ideal Sex Self-Dis	3.89	0.92	0.08	1.00	5.00	−1.15	1.23
Ought Sex Self-Dis	3.80	1.00	0.09	1.00	5.00	−1.06	0.62

**Table 3 healthcare-12-00798-t003:** Ought and ideal sexual traits by number of entries.

Ought Sexual Traits	*n*	Discrepancy	Ideal Sexual Traits	*n*	Discrepancy
Pain-Free	36	4.91	Pain-Free	59	4.2
Spontaneous	26	4.07	Carefree	35	3.82
Passionate	26	3.65	Spontaneous	29	4.03
Loving	25	2.4	Adventurous	23	3.86
Adventurous	20	4.04	Pleasurable	19	3.63
Sexy	19	3.5	Enjoyable	17	4.23
Easy	19	4.63	Sexy	17	3.94
Carefree	17	4.58	Playful	15	4.2
Enjoyable	17	3.94	Confidence	14	4.14
Fun	16	3.5	Loving	13	3.38
Seductive	15	4	Healthy	13	4.07
Arousing	15	3.73	Fun	12	3.75
Pleasurable	14	3.57	Passionate	12	3.91
Open-Minded	13	3	Romantic	11	3.18
Romantic	12	3.75	Seductive	10	4.3
Healthy	11	3.72	Sensual	10	3.3
Kinky	10	3.4	Easy	10	3.2
Playful	10	3.9	Excited	8	3.62
Flirty	9	3.44	Flirty	8	4
Amazing	9	3.77	Arousing	8	4.25
Engaging	9	3.88	Guilt Free	8	4.37
Shameless	9	3.88	Hot	8	4
Safe	9	2.66	Safe	8	3.5
Lustful	8	3.87	Amazing	5	4.6
Stimulating	8	3.375	Pressure-Free	5	2.8
Sensual	7	3.85			
Hot	7	3.57			
Naughty	6	4.5			
Guilt-Free	6	4			
Frequent	5	5			

**Table 4 healthcare-12-00798-t004:** Results for linear regression with ideal sexual self-discrepancy, ought sexual self-discrepancy, pain intensity, and frequency of sex, predicting sexual satisfaction.

Variable	*B*	SE	β	t	*p*
Ideal sexual self-discrepancy	−0.21	0.11	−0.18	−1.93	0.057
Ought sexual self-discrepancy	−0.14	0.10	−0.13	−1.36	0.176
Pain intensity	−0.33	0.15	−0.19	−2.11	0.031
Frequency of sex	0.24	0.07	0.30	3.29	0.001

## Data Availability

Data supporting reported results can be found at https://www.proquest.com/openview/aed6d09289e63f0908f26467424d3a30/1?pq-origsite=gscholar&cbl=18750&diss=y (accessed on 3 March 2024).

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
