# Peer review of "Sexual Self Discrepancies, Sexual Satisfaction, and Relationship Satisfaction in a Cross-Sectional Sample of Women Who Experience Chronic Vaginal Pain during Sexual Intercourse"

_healthcare, 2024, doi:10.3390/healthcare12070798_

Round 1

Reviewer 1 Report

Comments and Suggestions for Authors

First of all I would like to thank the editors for the opportunity to review the paper "Sexual Self Discrepancies, Sexual Satisfaction, and Relationship Satisfaction in Women Who Experience Chronic Vaginal Pain during Sexual Intercourse". I would like to congratulate the authors for their efforts. 

I would like to make some recommendations for improvement: 

Introduction. 

I recommend that you read and cite the paper "Sexual pain disorders" by Francisco Cabello-Santamaría, Francisco Javier del Río Olvera, and

Marina A. Cabello-García, published in the journal Current Opinion in Psychiatry (2015, 28:412-417), which reviews the different sexual pain disorders in women. 

Another important aspect, addressed by the same authors, is sexual pain disorders generated by drug use. I think it is important to include information in this regard, citing the paper "Sexual Pain Disorders in Spanish Women Drug Users", SUBSTANCE USE & MISUSE, 2017, VOL. 52, NO. 2, 145-151. 

Materials and Methods. 

It should be indicated how the authors think the extraneous variable of visa gift cards would affect. 

They should explain why they chose only a few scales and not the whole VPAQ questionnaire. The same applies to the GMSEX and GMREL scales. 

Information on the validity coefficient of the questionnaires should be included. 

Results. 

The authors state that the minimum number of participants is 82. Instead they use 106 participants. The sample size is calculated to avoid unnecessary cost and effort. The authors need to explain this discrepancy. 

Discussion. 

The authors report very little about their opinion of the different therapies that can help women overcome sexual pain disorder.

Author Response

Please see the attachment. Thank you for your time. 

Reviewer 2 Report

Comments and Suggestions for Authors

Dear Authors 

here my great congratulations for this work. it addresses a very hot topic, generally hard to define. methodology is great, data is consistent and their discussion are worthy to read. 

i would suggest to choice a more incisive title that needs to express the nature of the study (in this case, "cross sectional"). when i read the title i need to have study's main features well cleared . i would adjust abstract to make it more concise and punctual. regarding the conclusion i would ne more engaging, focusing more on the data expressed rather than what it needs to be. moreover, i would follow equator guidelines for writing and reporting cross-sectional study.
overall, the study deserves to be published.

Author Response

(The authors gave the same response as above.)

Reviewer 3 Report

Comments and Suggestions for Authors

The article is an important contribution to the literature as it sheds light on discrepancies between the ought and ideal sexual selves of women who experience pain related to vaginal intercourse. I have several comments for the authors to consider:

1.       The introduction gets a bit redundant at times and I thought could be tightened and reduced slightly with some editing.

2.       The authors note in the conclusions that the discrepancies described increase emotional distress, but emotional distress was not measured.

3.       The rationale for why there are discrepancies between ought and ideal self could be further expounded upon as this seems to be a significant contribution of this study.

4.       The authors also outline a number of steps that can be taken to minimize pain and maximize pleasure, but it is not clear where these recommendations come from and how they are derived from the results of this study.

Author Response

(The authors gave the same response as above.)
